# Dynamic Changes of a Thick Debris-Covered Glacier in the Southeastern Tibetan Plateau

Zhen He [1,2], Wei Yang [1,*], Yongjie Wang [1,3], Chuanxi Zhao [1,4], Shaoting Ren [1] and Chenhui Li [1]

1   State Key Laboratory of Tibetan Plateau Earth System, Environment and Resources (TPESER), Institute of Tibetan Plateau Research, Chinese Academy of Sciences, Beijing 100101, China
2   University of Chinese Academy of Sciences, Beijing 100049, China
3   South-East Tibetan Plateau Station for Integrated Observation and Research of Alpine Environment, Lulang 860119, China
4   College of Earth and Environmental Sciences, Lanzhou University, Lanzhou 730000, China
*   Correspondence: yangww@itpcas.ac.cn; Tel.: +86-010-8409-7049

**Abstract:** Debris-covered glaciers have contrasting melting mechanisms and climate response patterns if compared with debris-free glaciers and thus show a unique influence on the hydrological process. Based on high-resolution satellite images and unpiloted aerial vehicle surveys, this study investigated the dynamic changes of Zhuxi Glacier, a thick debris-covered glacier in the southeastern Tibetan Plateau. Our result shows that the whole glacier can be divided into the active regime and stagnant regime along the elevation of 3400 m a.s.l. The mean surface velocity of the active regime was 13.1 m yr$^{-1}$, which was five times higher than that of the stagnant regime. The surface-lowing rate of this debris-covered glacier reaches more than 1 m yr$^{-1}$ and displays an accelerating trend. The majority of ice loss concentrates around ice cliffs and supraglacial ponds, the ablation hotspots. These hotspots can be roughly classified into three types, including persistent, expanding, and shrinking patterns, at different dynamic regimes on the Zhuxi Glacier. With the evolution of these hotpots and glacier dynamic changes, the supraglacial ponds showed significant change, with the total number fluctuating from 15 to 38 and the total area increasing from 1128 m$^2$ to 95790 m$^2$ during the past decade. The recent exponential expansion of the proglacial lake and the significant downwasting of stagnant ice inside the dammed terminus moraine possibly trigger the glacial lake outburst flood and thus threaten the security of livelihoods and infrastructure downstream.

**Keywords:** debris-covered glacier; unpiloted aerial vehicle; ablation hotspot; moraine-dammed lake





## 1. Introduction

Debris-covered glaciers, which have extensive large rock debris covering most of the ablation area, are mainly distributed in Alaska, southwest Asia, and Greenland and are commonly found in high mountains [1,2]. The debris on these glaciers is mainly from the slope sliding on both sides of the glacier and the upwelling of inside moraine by ice flow [3]. Debris cover alters the glacier dynamics by changing surface albedo, thermal conductivity, and surface roughness [4–6]. The thin debris cover can enhance the surface melting by absorbing more energy. When the debris thickness is greater than a certain thickness (~10–30 mm), the debris will block the conduction of surface heat, thereby inhibiting the underlying melting [7,8]. Compared with debris-free glaciers, such debris-covered glaciers have distinct characteristics, including ablation rates, surface topography, and response mechanisms to climate change [9–11].

Due to the heterogeneous sub-debris ablation, supraglacial lakes and ice cliffs are, therefore, popularly developed on debris-covered glaciers [12]. These landforms responded rapidly to the ablation and terminus retreat [13–16]. The ice cliffs and supraglacial lakes generally have larger melt rate than those with thick debris, acting as ablation hotspots [17–19]. The ice cliff area of the debris-covered Ngozumpa glacier in the south slope of Mt Everest

accounts for about only 5% of the total glacier area, but the mass loss contribution from ice cliffs reaches to as high as 40% of the total [20]. Generally, the ice cliff and supraglacial ponds spatially co-exist. Approximately 77% supraglacial lakes have adjacent ice cliff one the debris-covered glaciers in Mt. Everest [21].

Ice cliffs and supraglacial ponds change dynamically on debris-covered glaciers. They usually retreat a long distance or have significant volume change within a few weeks or months [20,22]. The mass loss and lifespan of an ice cliff is largely depending on its aspect. In the Northern Hemisphere, south-facing ice cliffs tend to rapidly melt to disappear within a few weeks due to more solar shortwave radiation receipts, while less melt for the north-facing ice cliffs and thus retreat persistently [23]. During the ablation season, the supraglacial ponds are replenished by meltwater, thereby rapidly expanding in area [24]. These supraglacial lakes may also drain rapidly within a few weeks by connecting englacial conduits [14]. The rapid retreat of debris-covered glaciers often results in the formation or expanding of moraine-dammed proglacial lakes at its terminus [25] and thus pose the risk of glacier lake outburst floods (GLOF).

High Mountain Asia (HMA) is the most concentrated area of debris-covered glaciers [26,27], especially in the Himalayas and southeast Tibetan Plateau (TP). Previous research is mostly concentrated on the south slope of Mt. Everest region in the Himalayas [1,12,14,20,28,29]. The southeastern TP is another concentration regime of debris-covered glaciers in HMA [27,30], but few studies focus on the dynamic change of debris-covered glaciers [31,32]. Due to the abundant precipitation from the Indian summer monsoon and high mountain topography, the southeastern TP has a large number of temperate glaciers with high accumulation and ablation, and 16.9% glacierized area is covered by debris [27]. Temperate glaciers in this region are very sensitive to climate change and have suffered from significant ice loss in the recent two decades [33–35]. The presence of debris cover, ice cliffs, and supraglacial lakes complicated the climate response of temperate glaciers in this region [19]. The variation of surface elevation and surface velocity affects the development of ablation hotspots. Meanwhile, different kinds of ablation hotspots further control the morphological changes of the debris-covered glacier. Some ablation hotspots possibly turn into proglacial lakes with the potential risk of GLOF. The study on glacier dynamic changes by using high-resolution data, especially for the evolution of ablation hotspots and supraglacial lakes, is essential for understanding the distinct response of debris-covered glaciers to climate change in the southeastern TP.

High-resolution unpiloted aerial vehicle (UAV) systems are being increasingly used in glaciological studies worldwide, partly because they can overcome many of the shortcomings associated with both satellite remote sensing and in situ measurement. Compared with conventional in situ field measurement, the UAV flying technique is more flexible and low-cost. In addition, compared with satellite remote sensing, the images obtained by UAV have the advantages of high-resolution, free acquisition time, and friendly weather requirements [17,36,37]. Based on the images taken by UAV, the Structure from Motion (SfM) technique is used to generate high-resolution digital orthophoto maps (DOMs) and digital surface models (DSMs) [38]. This method has popularly been used in glaciological research, particularly for debris-covered glaciers [22,23,28,39–41].

In this study, different datasets including satellite images and UAV surveys are used to investigate the dynamic changes of Zhuxi Glacier, which is a thick debris-covered glacier in the southeastern TP with an expanding proglacial lake and is closed to the dense population and important infrastructure downstream (e.g., the Sichuan–Tibet road and planned railway). The main goals of this research are (a) to clarify the spatial pattern of glacier dynamics including surface elevation and surface velocity, (b) to quantify the development of ablation hotspots, (c) to discuss the possible risk of the expanding proglacial moraine-dammed lake in the Zhuxi Glacier.

## 2. Study Site

The Zhuxi Glacier (29°59′N, 95°30′E) was located in the southeastern TP, where 8.3% of glacierized area are covered by debris and the mean debris thickness is approximately 0.18 m [42]. Zhuxi Glacier is a thick debris-covered valley-type glacier with an area of about 15 km$^2$. It flows northeast from 5236 m a.s.l. to 3133 m a.s.l. at its terminus (Figure 1). The glacier tongue below 3800 m a.s.l. is covered by thick debris, accounting for about 37.5% of total glacierized area. The debris-covered tongue has a gentle slope about 4.3 degrees. The upper part above 3800 m a.s.l. is accumulation zone with the supply from frequent snow/ice avalanches. The slope rises to about 30 degrees in upper part (>3800 m a.s.l.), and the coverage of debris decrease to bare ice with altitude. Compared with other nearby debris-covered glaciers, e.g., 24 K Glacier and Hailuogou Glacier [10,43,44], the debris on Zhuxi Glacier is thicker with the mean value of more than 2 m (observed from the exposed debris layer). Due to the thick debris cover, the surface of Zhuxi Glacier is rugged accompanied by many ice cliffs and supraglacial ponds, which accounts for about 0.5% total area of debris-covered region. Some shrubs are developed on thick debris-covered area. Compared with other debris-covered glaciers in the south-slope Himalayas [1,12,14,20,28,29], Zhuxi Glacier have much lower elevation, higher air temperature [32]. According to the automatic weather station located in Zhuxi village (Figure 1), the mean annual temperature and annual total precipitation were 8.4 °C and 768 mm, respectively.

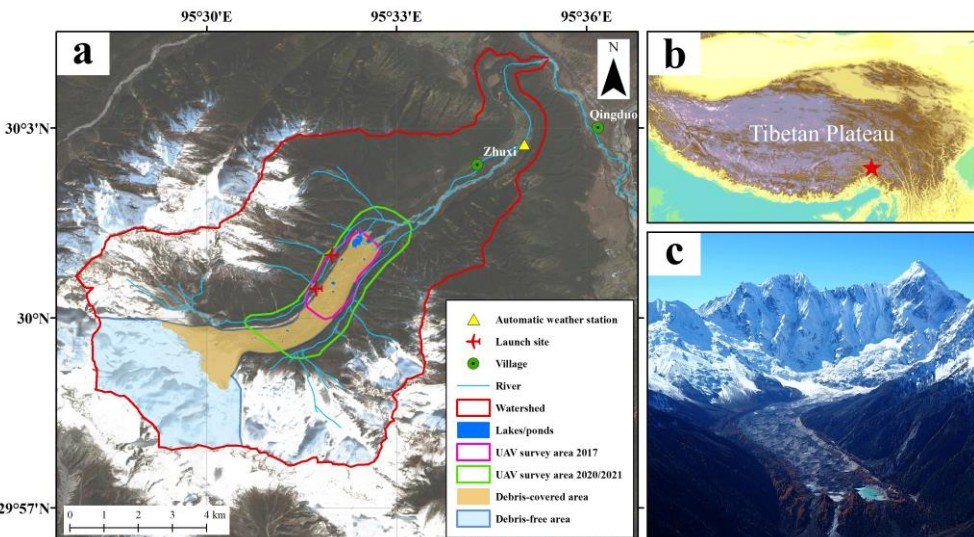

**Figure 1.** (**a**,**b**) The location of Zhuxi Glacier with the extent of the UAV survey in 2017 and 2020/2021, the distribution of debris-covered and debris-free areas, and supraglacial ponds, as well as the rivers, contours, watershed, and villages nearby. The background image in (**a**) is a Planet 4-band image acquired in 2020. (**c**) The panorama photo taken in 2020.

## 3. Data and Method

### 3.1. Surface Elevation Changes Based on Satellite-Derived DEM Difference

In this study, the comparison between the Shuttle Radar Topography Mission (SRTM) Digital Elevation Model (DEM) acquired in 2000 and the ZY-3 DEM in 2016 is used to derive the surface elevation change of Zhuxi Glacier during the period of 2000–2016. The SRTM DEM is a 30 × 30 m resolution global DEM dataset generated from the C-band of SRTM. These data were acquired by the United States Geological Survey (USGS) and the German Aerospace Center (DLR) in February 2000. The comparing DEM has been generated from the ZiYuan-3 (ZY-3) three-line-array (TLA) stereo images. The ZY-3 TLA is derived from the ZY-3 satellite launched by China in January 2012. Each ZY-3 TLA stereo image includes Nadir, Backward and Forward images with a resolution of 2.5 m, 3.5 m, and 3.5 m, respectively. Using the Imagine Photogrammetry tool of ERDAS IMAGINE 14 software, high-precision DEM with a resolution of up to 10 m in this area is generated

based on the above TLA images [45]. To generate reliable DEM, tie points of N, B, and F images, horizontal control points based on locations of features with Google Earth, and vertical control points based on the altitude of features with SRTM need to be added. The number and root mean square error (RMSE) of each tie point, horizontal control point, and the vertical control point is shown in Table 1.

**Table 1.** The number of tie and control points with RMSE in DEM generation process of ZY-3 TLA.

| Points | Number | RMSE/m | | |
|:---:|:---:|:---:|:---:|:---:|
| | | x | y | z |
| Tie points | 35 | 0.18 | 0.21 | |
| Horizontal control points | 17 | 0.45 | 0.57 | |
| Vertical control points | 641 | | | 1.90 |

The generated DEM has a large vertical error in steep slope area. The reason is mainly due to the shadows and snow coverage on mountains, which makes the tie points and horizontal control points mostly located in plains of valleys where have no disturbance of shadows and snow instead of steep slope mountain area. To further improve the accuracy in steep slope area, this research filters out the areas where the height deviation between the ZY-3 DEM and SRTM DEM exceeds 100 m and perform co-registration where the error is less than 100 m. This co-registration utilizes the cosine relationship between the elevation difference and the terrain slope and aspect caused by the horizontal coordinate offsets [46] and corrects these offsets according to the distribution of the elevation difference by using the demcoreg tool [47].

### 3.2. Surface Elevation Changes based on UAV-Derived DEM Difference

To extract the high-spatial-resolution DOMs and DEMs, UAV surveys were performed three times in December 2017, September 2020, and October 2021, respectively. The eBee Plus fixed-wing UAV system was used in the 2017 flight. This UAV has a built-in Real-Time Kinematic (RTK) technique, with a manufacturer-stated horizontal accuracy of 3 cm and vertical accuracy of 5 cm. The flight time of this survey is selected as 10:00–11:00 when the sky is clear and there is no fresh snow and obvious shadows on the glacier surface. The 2017 UAV survey took 476 photos for the area covering approximately 1.5 km² below 3400 m a.s.l. This high-resolution survey is mainly used for investigating the dynamic change of ice cliffs and supraglacial ponds in stagnant regimes.

On September 2020 and October 2021, DJI Phantom 4 RTK was chosen to conduct the UAV survey. This UAV also has built-in RTK with higher manufacturer-stated accuracy than eBee Plus (horizontal accuracy of 1 cm and vertical accuracy of 1.5 cm). The flight time in 2020 was 15:00–18:00, and total of 343 photos were taken, while the flight time in 2021 was from 11:00 to 15:00, and a total of 380 photos was taken. Due to the larger survey area and rich vegetation nearby, the signal transferring quality was poor when the drone was far from the launch site. To solve this issue, two fixed launch sites are set up to cover the upper and lower part of survey area during these two flights (Figure 1).

The images obtained by UAV surveys were used as inputs for the SfM reconstruction using the pix4Dmapper software (v. 4.3.31) (Prilly, Switzerland). This method switches hundreds of images into high-precision optical DOMs, then calculates high-precision point cloud data and DEM [38]. Due to the different UAV systems and flight altitudes, the spatial resolutions of DOMs and DEMs are different. The resolution of DOM and DEM derived from eBee Plus in 2017 is 0.06 m, while the 2020 and 2021 flights derived a spatial resolution of about 0.15 m.

Due to the shift of RTK base station, the upper part and lower part of DOMs and UAV-derived DEMs of 2020 and 2021 flight need to process individually. The overlapping area between the upper and lower areas ware used as the co-registration area. By aligning the selected ground control points (GCPs) on the DOMs, the horizontal errors of DOMs and

UAV-derived DEMs in the upper and lower areas were eliminated. After the horizontal correction, the vertical difference of the upper and lower DEMs were eliminated by minimizing the elevation difference between the upper and lower overlapping co-registration area.

The co-registration process of DOMs and UAV-derived DEMs in different years is similar to the above-mentioned co-registration of upper and lower parts of the flights in 2020 and 2021. The differences are the selection of GCPs and co-registration area. All the GCPs are boulders or bedrock located in off-glacier area. A total of 14 selected virtual GCPs was used to minimum the different horizontal and vertical errors between three UAV surveys. The calibration processes are implemented by using ArcGIS 10.5 software. The flat rubble beach on the north of the glacier terminus and the grassland on both sides of the glacier (Figure 1) were chosen as off-glacier co-registration areas, these areas show no change in surface features between 2017 and 2021.

### 3.3. The Calculation of Surface Velocity

The surface velocity of Zhuxi Glacier was calculated from the PlanetScope One Satellite 4-band images during the period of 2016–2020 and UAV-derived DOMs during the period of 2020–2021 by using the Co-registration of Optically Sensed Images and Correlation (COSI-Corr) tool [48]. The COSI-Corr is a software package in Environment for Visualizing Images (ENVI) software, which can calculate the displacement on remote-sensing images or UAV-derived DOMs by feature tracking. The COSI-Corr tool has been used widely on glacier surface velocity calculation [17,49–51]. The COSI-Corr tool generated three images, which include the north–south displacement, east–west displacement, and signal–noise ratio. The area where the signal-to-noise ratio is less than 0.9 and outliers were removed. The blank area was filled by interpolation. The annual surface velocity results during 2016–2021 were obtained by the combination of several annual displacement images of north-south and east-west.

### 3.4. The Extraction Process of Supraglacial Lakes

The Rapideye images in 2009–2015 and the PlanetScope One Satellite 4-band images in 2016–2020 were used to extract the glacial lake area from 2009 to 2020 (Table 2). The images are collected in October or November, with a cloud fraction of less than 5% and less snow cover on the glacier surface. The normalized difference water index (NDWI) [52] uses the ratio of the near-infrared band to the green band on remote-sensing images, which can significantly distinguish water and non-water bodies. This index has been used in supraglacial lakes area extraction in the south-slope Himalayas [53–55]. In this study, the boundary of the supraglacial lakes was extracted by adjusting the threshold of the NDWI index. Due to the different atmospheric radiation conditions of remote sensing images at different times, the NDWI extraction threshold of each image is different, generally ranging between 0.1 and 0.25. Manual inspection was performed to modify inaccurate lake boundary. The supraglacial lakes with an area of less than 9 m$^2$ were removed.

**Table 2.** Datasets and the relevant information used for the Zhuxi Glacier.

| Purpose | Product Type | Acquisition Date | Resolution |
|---|---|---|---|
| Surface elevation change | C-band Digital Elevation Model of SRTM | February 2000 | 30 m |
| | ZY-3 TLA stereo images | October 2016 | Nadir: 2.5 m<br>Forward: 3.5 m<br>Backward: 3.5 m |

**Table 2.** *Cont.*

| Purpose | Product Type | Acquisition Date | Resolution |
|---|---|---|---|
| Lake area extraction | PlanetScope Rapideye | November 2009<br>November 2010<br>November 2011<br>November 2012<br>October 2013<br>November 2015 | 5 m |
| Lake area extraction<br>Surface velocity | PlanetScope One Satellite 4-band | November 2016<br>November 2017<br>November 2018<br>November 2019<br>November 2020 | 3 m |
| Surface elevation change<br>Lake and ice cliff change | eBee Plus survey | December 2017 | 0.06 m |
| | DJI RTK surveys | September 2020<br>October 2021 | 0.15 m |

*3.5. Uncertainty*

The accuracy of glacier surface elevation change was estimated by calculating the elevation difference in stable off-glacier areas. Figure 2 shows the histograms of vertical errors of surface elevation change in 2000–2016 and 2020–2021 with their mean value and RMSE. The vertical uncertainties of surface elevation change during 2000–2016 mostly originate from the DEM extraction process from ZY-3 TLA. The uncertainty of surface elevation change during 2020–2021 is mainly from outliers originating from the vegetation SfM calculation. The mean value of elevation difference in the stable off-glacier area with normal distribution by UAV survey is 0.02 m, lower than the elevation difference in the off-glacier area by satellite images for ~0.05 m.

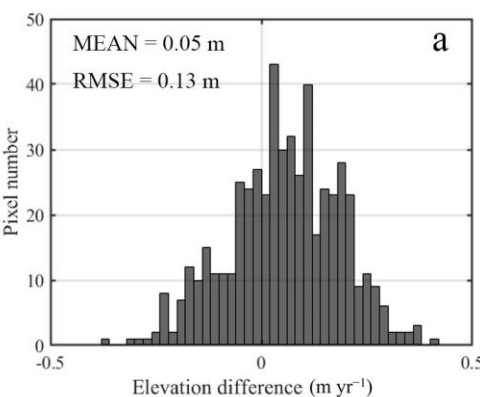
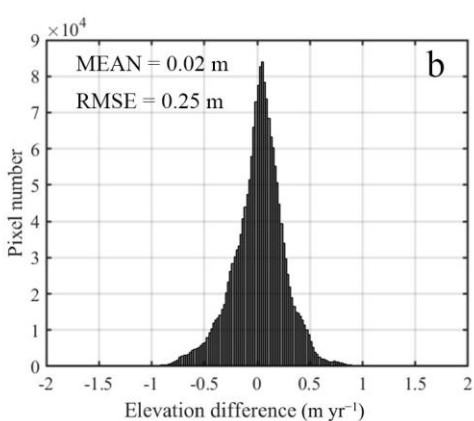

**Figure 2.** The deviation histogram of vertical errors of surface elevation change in 2000–2016 (**a**) and in 2020–2021 (**b**) over the off-glacier area.

The off-glacier area of 2020–2021 surface elevation difference result has higher RMSE (~0.25 m) than that of the 2000–2016 result (~0.13 m). The much higher resolution of UAV image (0.06 m, 0.15 m) enlarged the effect of outliers, which exist on the edge of trees and shrubs. These outliers were filtered in 2000–2016 result under the spatial resolution of 30 m. Though having larger RMSE, 94.6% vertical errors of elevation change in 2020–2021 lie within the range of $-0.5{\sim}0.5$ m yr$^{-1}$, which is accurate enough for the results below.

The uncertainties in lake area extraction mainly comes from the relatively low resolution of Planet 4-bands images (3 m) and the selection of NDWI threshold, which significantly affects the extracted lake area. The weather and atmospheric radiation states vary from image to image, resulting in different NDWI thresholds for extracting lake boundaries in

each image. The accurate NDWI values corresponding to the lake boundaries needs to be distinguished with the help of optical images. Meanwhile, the NDWI values of the ice cliffs next to the glacial lake sometimes fall within the threshold and are mixed with the shallower water bodies on the lake shore. It is necessary to separate the cliff and lake with the help of true-color images and clip them manually. Similar to the previous study [55], the uncertainty is determined by selecting 0.5 pixels boundary of the extracted lake area.

## 4. Results

### 4.1. Spatial Patterns of Surface Elevation Change

Figure 3a shows the spatial pattern of mean annual surface elevation changes of Zhuxi Glacier and their mean altitudinal variation during the period from 2000 to 2016. The surface elevation change is characterized by a heterogeneous spatial pattern, with several ablation hotspots (the mean elevation decreases of more than 3 m $yr^{-1}$) in the middle section of Zhuxi Glacier. The limited change mainly occurs at the glacier terminus where was covered by thicker debris, and the mass loss gradually became larger in the middle zone of ablation area, and then shift to slight mass gain in upper part above the elevation of 3800 m where were supplied by frequent snow avalanches. The average surface elevation change was $-0.93 \pm 0.13$ m $yr^{-1}$ during the period from 2000 to 2016.

Figure 3b shows the surface elevation change of Zhuxi Glacier during 2020–2021 based on UAV-derived DEMs comparisons. In contrast to the satellite-derived results (Figure 3a), the UAV-derived DEM difference provided the high-resolution result for investigating the detailed surface process. For the region above than 3400 m a.s.l., there showed alternating pattern of positive and negative surface elevation change with large standard deviation values (Figure 3b). For the region below 3400 m a.s.l., the thick debris inhibit the underneath ice melting and substantial surface elevation change mainly occurred in ablation hotspots characterized by the cliff-pond systems. The area with an surface elevation change exceeding $-3$ m $yr^{-1}$ (ablation hotspot regions) only account for about 14% of the area below 3400 m a.s.l., but the mass loss of which account for about 92.4% of total. The average surface elevation change of ablation hotspot regions was up to $-6.7$ m $yr^{-1}$, which is 6.6 times higher of $-1.01 \pm 0.38$ m $yr^{-1}$ for the total area below 3400 m a.s.l. Two representative area (Area 1 for thick debris-covered area, Area 2 for significant surface elevation change area) were selected to show their differences on surface elevation change (Figure 3d). The average surface elevation change of Area 1 was $-0.20$ m $yr^{-1}$ whereas Area 2 was $-3.07$ m $yr^{-1}$ in 2020–2021. The largest surface elevation change can reach to $-20$ m $yr^{-1}$ in Area 2. The averaged value in UAV survey area was $-1.47 \pm 0.25$ m $yr^{-1}$, which was higher than that of $-0.99 \pm 0.13$ m $yr^{-1}$ of satellited-derived DEM difference for the corresponding area during the period of 2000–2016. Apparently, it showed an accelerated mass loss trend even for this thick debris-covered glacier in the southeastern TP.

### 4.2. Spatial Distribution of Surface Velocity

The satellite-derived surface velocity of Zhuxi Glacier showed significant spatial heterogeneity (Figure 4). According to the isopleth with a surface velocity of 7 m $yr^{-1}$, the glacier can be divided into an active regime in the upper part and a stagnant regime in the lower part along the elevation of ~3400 m a.s.l. During 2016–2020, the active regime was characterized by the dynamic flow with a surface velocity ranging from 7 to 20 m $yr^{-1}$. In contrast, the surface velocity in the stagnant regime decreased to the values of 0.6 to 7 m $yr^{-1}$, which indicates the stagnant condition in this thick debris-covered zone (Figure 4a). The mean surface velocity in the active regime (13.1 m $yr^{-1}$) was about five times higher than that of the stagnant regime (2.5 m $yr^{-1}$). The altitudinal pattern of surface velocity shows an increasing trend, and the maximum surface velocity (19.6 m $yr^{-1}$) is concentrated in the elevation of approximately 3450 m a.s.l., which is 2.8 km from the glacier terminus.

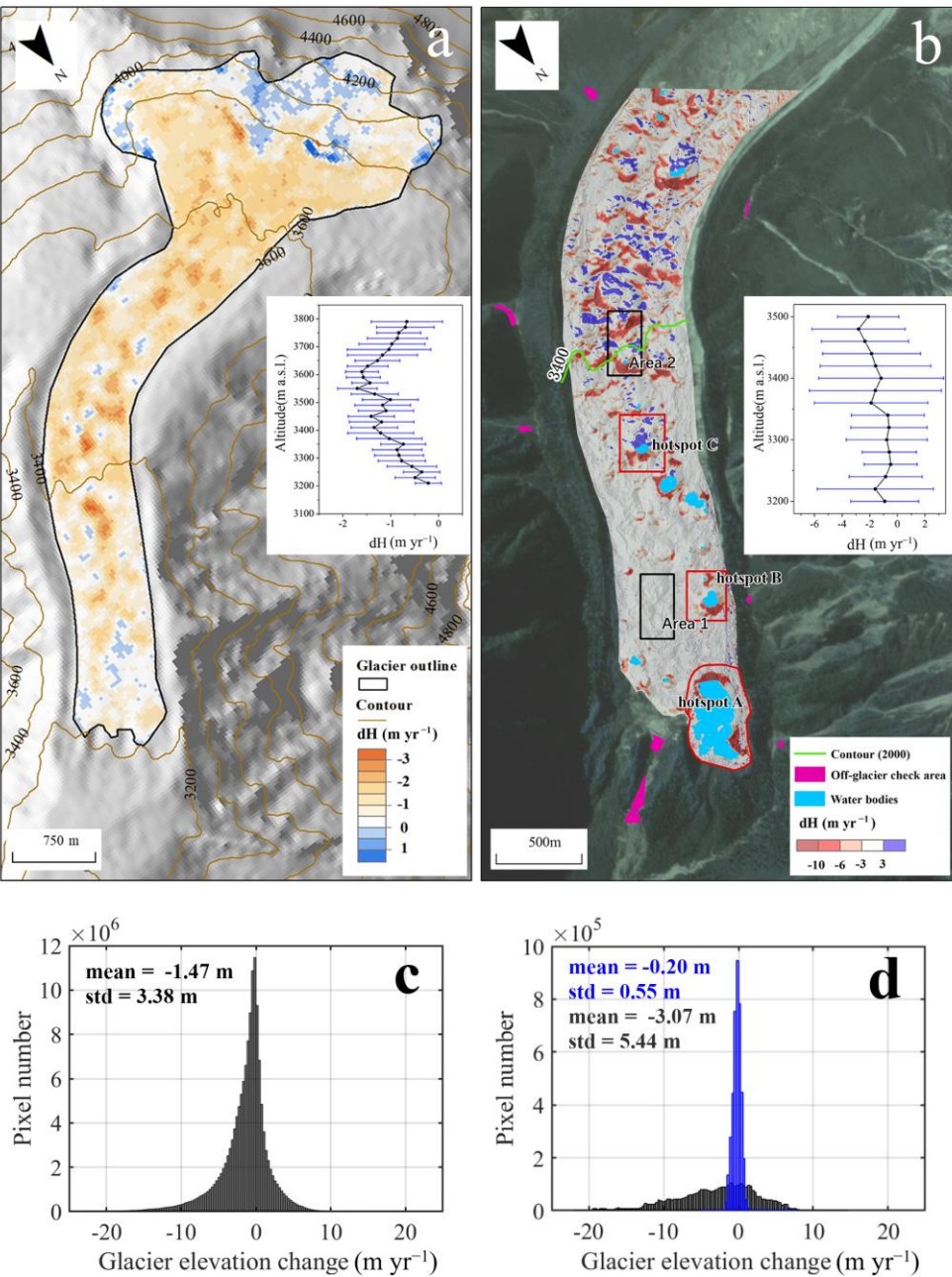

**Figure 3.** The mean surface elevation change in 2016–2020 (**a**) and in 2020–2021 (**b**), together with the mean glacier surface elevation change within 20 m elevation bands (black dots) with the corresponding standard deviation (blue horizontal bar). (**c**) The deviation histogram of surface elevation change in 2020–2021. (**d**) The deviation histogram of surface elevation change over two representative regions: Area 1 with thick debris cover (blue) and Area 2 with significant surface elevation change (gray) in 2020–2021. The locations of Area 1 and Area 2 are shown in (**b**).

The surface velocity derived from UAV-derived DOMs during 2020–2021 shows a similar spatial pattern to the satellite-derived velocity (Figure 4b). The active regime extended more down to the altitude of ~3350 m a.s.l. from ~3400 m a.s.l. in 2016–2020. The maximum surface velocity (17.8 m yr$^{-1}$) is located at the elevation of ~3420 m and 2.4 km from the glacier terminus. The mean (7.7 m yr$^{-1}$) and the maximum (17.8 m yr$^{-1}$) in the UAV surveying region were slightly reduced if comparing the corresponding satellite-derived velocity (mean and maximum of 7.8 and 19.6 m yr$^{-1}$).

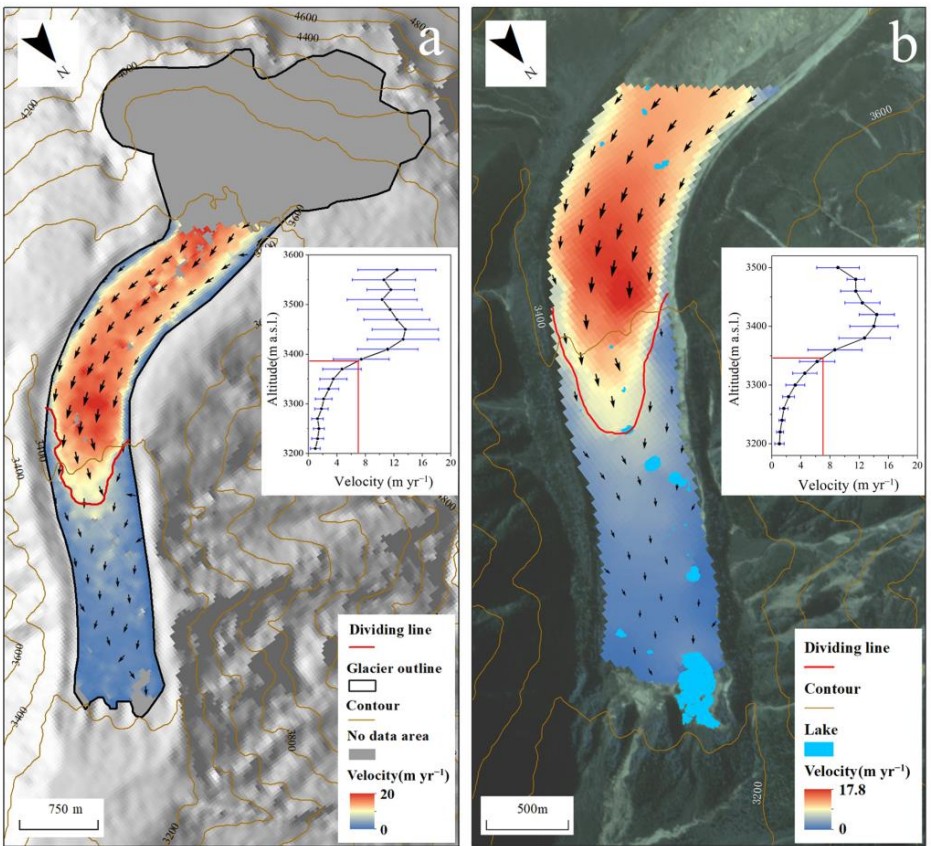

**Figure 4.** Spatial pattern of surface velocity of Zhuxi Glacier in 2016–2020 (**a**) and in 2020–2021 (**b**), together with their mean surface velocity changes within 20-m elevation bands (black dots) with standard deviation (blue horizontal bars) in each inset panel. The red beelines indicate the position of an average surface velocity of 7 m yr$^{-1}$ and the corresponding altitude. The dividing line showed in red is the isopleth with a surface velocity of 7 m yr$^{-1}$.

### 4.3. The Evolution of Water Bodies

The water bodies of Zhuxi Glacier consist of several small supraglacial ponds and a large moraine-dammed lake at the glacier terminus (Figure 5a). The supraglacial ponds are widely distributed in the debris-covered region, ranging from 3800 m a.s.l. to the terminus. During 2009–2020, the number of supraglacial ponds displayed a large inter-annual fluctuation, with a minimum number of 15 in 2015 and a maximum of 38 in 2018 (Figure 5b), partly due to the dynamic influence of pond formation and drainage in the active regimes. The water bodies underwent a significant exponential growth at an average rate of 13.4% yr$^{-1}$, with a total area of ~3 × 10$^4$ m$^2$ in 2009 but ~9 × 10$^4$ m$^2$ in 2020 (Figure 5b). In active regimes (above 3400 m a.s.l.), both the number and area of supraglacial ponds show less change in the past decade. In contrast, due to the significant area expansion of the moraine-dammed lake, the total water bodies in the stagnant regime (below 3400 m a.s.l.) experienced a notably increasing (Figure 5c).

The moraine-dammed lake at the terminus was initially a small supraglacial pond with a total area of 1128 m$^2$ in 2009, then expanded rapidly to 20,309 m$^2$ in 2016 and to 95,790 m$^2$ in 2021 (Figure 6), which was 85 times the area in 2009 with an exponential growth rate of 44.8% yr$^{-1}$. It should be noted that the meltwater stream channel migrated from the east side of the terminus to the west side in 2017 due to the glacier retreat, which caused the meltwater to flow into the moraine-dammed lake and significantly enlarged the lake area since 2018. The accelerated expansion was also contributed by the upstream retreat of the ice cliff and the downward meltdown of stagnant ice. The detailed process of lake expansion and its influence will be investigated in the Discussion section.

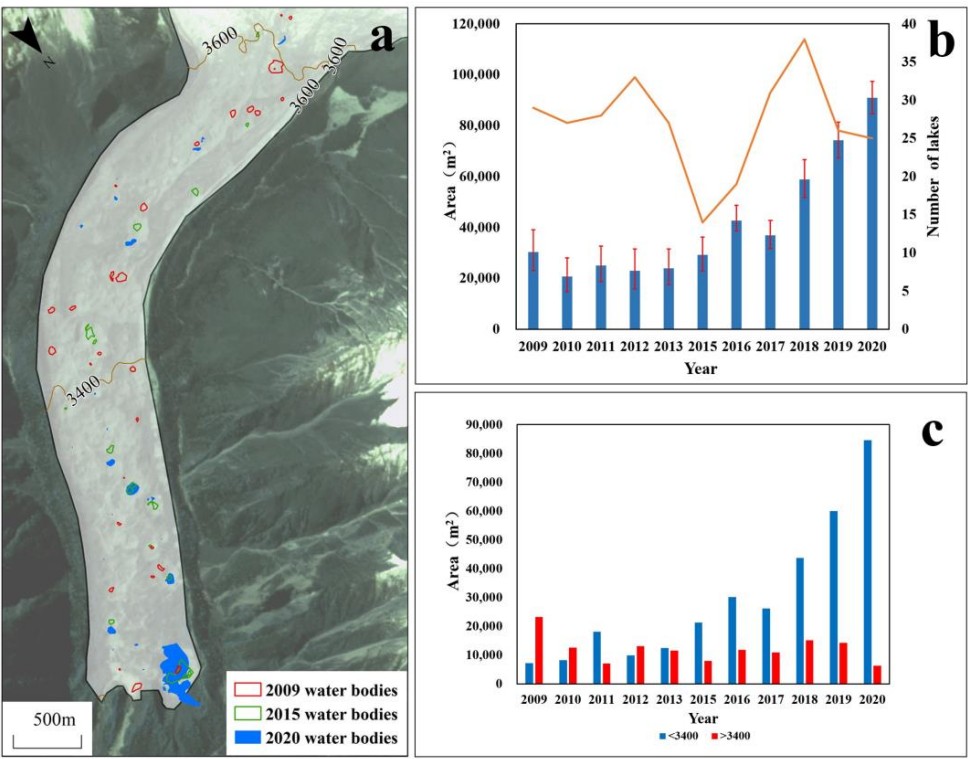

**Figure 5.** (**a**) The spatial distribution of supraglacial ponds, moraine-dammed lake in 2009, 2015, 2020. (**b**) The number (yellow line) and the total area (blue bars) with possible errors of ponds and lakes. (**c**) The evolution of water bodies in active regime (above 3400 m a.s.l.) and stagnant regime (below 3400 m a.s.l.).

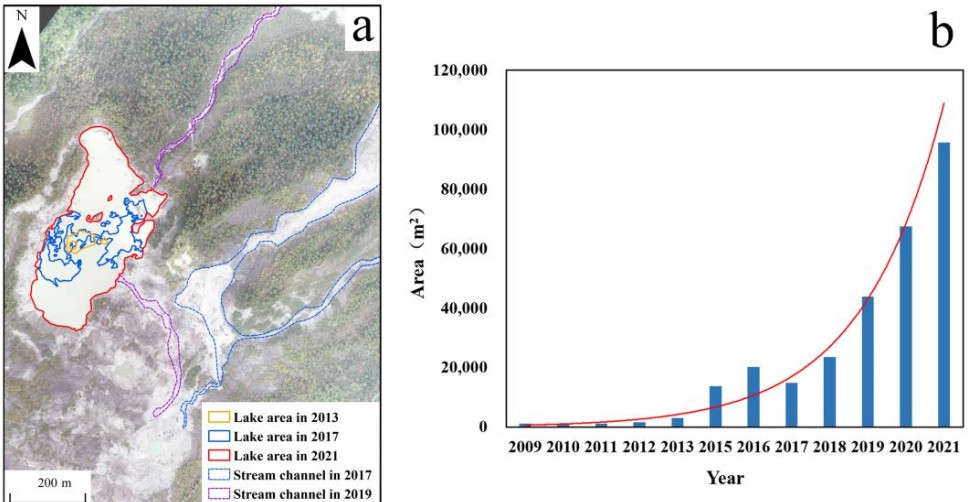

**Figure 6.** (**a**) The evolution of the moraine-dammed lake at the terminus in 2013, 2017, and 2021, together with the migration of the stream channel in 2017 and 2019. The background image is DOM acquired in 2021. (**b**) The annual lake area and its exponential increasing trend (red line) during the period between 2009 and 2021.

## 5. Discussion

### 5.1. The Dynamic Change of Different Type Cliff-Pond Systems

The ablation hotspots in Zhuxi Glacier mainly consists of ice cliffs and supraglacial ponds, called the cliff-pond systems [14,56]. According to their dynamic evolution and geographical locations, three representative types of cliff-pond/lake systems were roughly classified on the whole glacier. These types were the persistent cliff-lake system at glacier

terminus (named as hotspot A), the expanding cliff-pond system at stagnant regime (named as hotspot B), and the shrinking cliff-pond system at active regime (named as hotspot C), respectively (Figure 7). The locations of these three representative types are shown in Figure 4a.

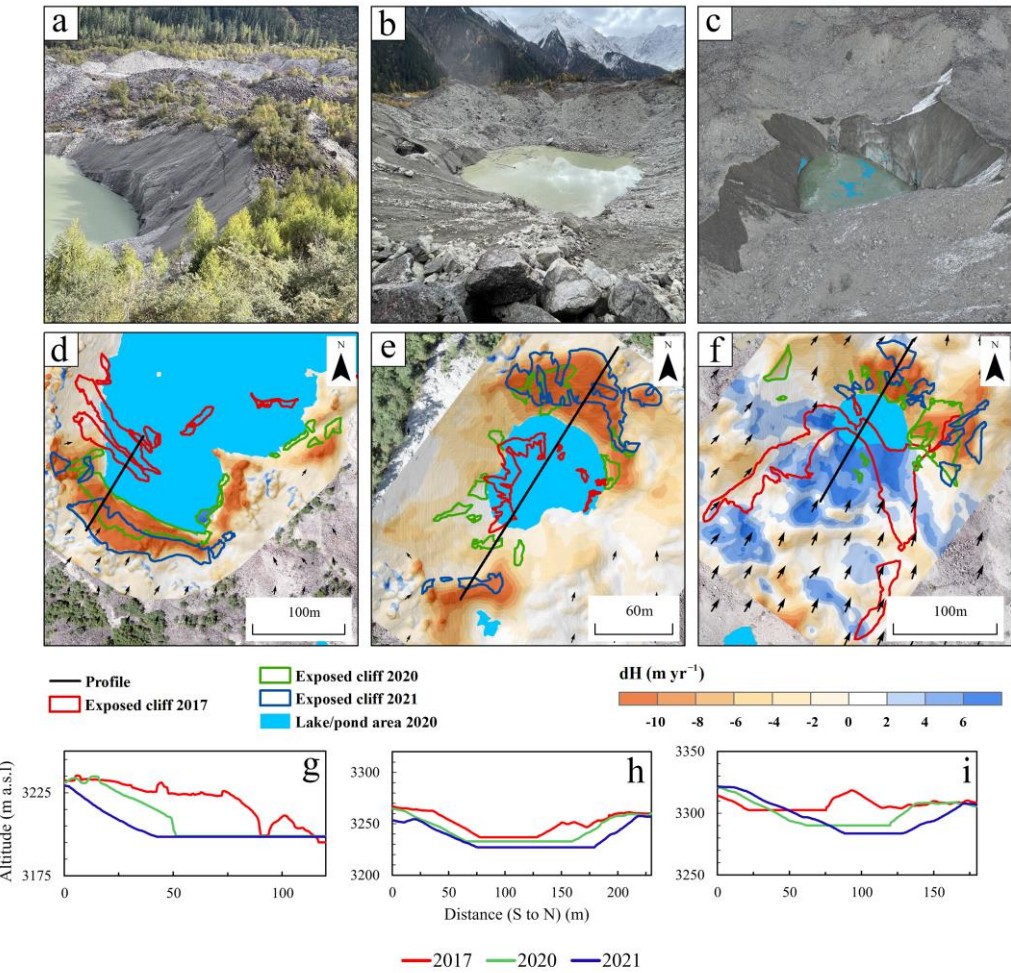

**Figure 7.** (**a**,**b**) The photos of hotspot A and hotspot B, taken in October 2021. (**c**) The point cloud model of hotspot C originate from UAV flight in December 2017. (**d**–**f**) Dynamic change of three different hotspot system showing the surface elevation change during 2020–2021, the evolution of ice cliff, the surface flow vectors, together with the selected topographic profiles during 2017, 2020 and 2021 along hotspot A (**g**), hotspot B (**h**) and hotspot C (**i**). The background image is from the DOM in 2020.

The hotspot A was a persistent cliff-lake system at the glacier terminus characterized by a north-facing steep ice cliff and an adjacent large moraine-dammed lake (Figure 7a). The hotspot A was similar to those hotspots defined as the calving termini in the Himalayas, which play an important role in the rapid expansion of proglacial lakes [57–59]. Under the calving effect of such proglacial lakes, the ice cliffs retreated significantly [60]. As shown in Figure 7g, the ice cliff had retreated ~84 m during the past four years, with a total of ~64 m retreat in 2017–2020 and ~20 m in 2020–2021. The long-distance retreating caused considerable surface elevation change for about ~$-15$ m yr$^{-1}$. The corresponding retreated area was occupied by the lake and therefore expanded the proglacial lake southward. The calving process consistently changed the geometry of the ice cliff. In 2017, the north part of a separate ice cliff was split from one complete ice cliff (Figure 7d). When the submerged part of the ice cliff was carved by the lake, the unsupported ice cliff above therefore collapsed by expanding the crevasses on the complete ice cliff [60]. This event increased the slope of

the ice cliff but shrunk the ice cliff area. The collapsed ice cliff is converted to floating ice, and the rest of the ice cliff will gradually flatten until it collapses again. In 2020–2021, the ice cliff flattened gradually from a slope of 42.5° in 2020 to 39° in 2021. In contrast, the cliff area increased gradually from 3661 m$^2$ in 2020 to 4483 m$^2$ in 2021. The sudden collapse of a 30 m high ice cliff at the Zhuxi Glacier may create a seiche or displacement wave on the proglacial lake, which may destabilize or overtop the ice-cored dam. Such a proglacial cliff-lake system acts as an important potential trigger for GLOF [61,62].

The hotspot B is an expanding circular cliff-pond system located at a stagnant regime (Figure 7b), characterized by low surface velocity for 2–3 m yr$^{-1}$. A similar circular cliff-pond system is commonly found in debris-covered glaciers in the Himalayas [56,63]. The total exposed area of the ice cliff in hotspot B increased rapidly from 1258 m$^2$ in 2017 to 1705 m$^2$ in 2020, then 3320 m$^2$ in 2021 (Figure 7e), while the slope decreased slightly from 35.2° in 2017 to 32.8° in 2020 and 2021. The pond area expanded from ~2470 m$^2$ in 2017 to ~5118 m$^2$ in 2021 with the retreating of ice cliffs. The ice cliff retreated in all directions, with the most rapid retreat in the south-facing part for a total of about 52 m retreat in 2017–2021 but the less ice cliff retreat in the north-facing part with a total retreat of 14 m (Figure 7h). The cliff retreat caused the corresponding negative surface elevation change of ~15 m in the south-facing part and ~3 m in the north-facing part. This contrasting change is partly related to the different aspects of ice cliffs receiving solar shortwave radiation and partly related to the redistribution of debris cover. In Northern Hemisphere, the south-facing ice cliffs have enhanced solar radiation receipt, which promoted the backwasting and flattening process [23,56]. Moreover, under the effect of relatively low slopes and frequent rockfall, the redistribution of debris-covered ice cliff areas also substantially influenced the backwasting of ice cliffs [20,28]. In 2020–2021, the south-facing ice cliff was largely exposed, accompanied by the remarkable backwasting of 30 m. The exposed area in east-facing and north-facing was reburied; hence, little backwasting and negative surface elevation change occurred in 2020–2021. The evolution of such kind of cliff-pond system is possibly controlled by the cliff aspects and their slope change.

The hotspot C was located in the active regime of Zhuxi Glacier, which showed a shrinking and flattening pattern (Figure 7f). In 2017, this cliff-pond system consisted of a large and steep south-facing ice cliff and an adjacent supraglacial pond (Figure 7c). However, the cliff has experienced significant flattening accompanied by the reburying and backwasting process since 2017. The slope of the ice cliff decreased significantly from 44.5° in 2017 to 30° in 2021 (Figure 7i). The slope of 30° is generally assumed to be a threshold of an ice cliff that be reburied with debris [64]. The flattening of the ice cliff in hotspot C favored the debris reburying process, and the total area of exposed ice cliff area was only 2140 m$^2$ in 2021, which was 74.3% smaller than the area of 8318 m$^2$ in 2017 (Figure 7f). The area of the adjacent pond shrunk sharply from 2890 m$^2$ in 2017 to 1483 m$^2$ in 2021. The water level of this pond also showed a substantial decrease of about 20 m in 2017–2021 (Figure 7i). The high ice flux was the main controller of the dynamic change of hotspot C. Unlike hotspots A and B, the region of hotspot C was relatively active with higher surface flow velocity (8 m yr$^{-1}$ in upstream and 5.5 m yr$^{-1}$ downstream). For the upper part of hotspot C, the more ice flux input and inclined slope led to the positive surface elevation change for about 2~10 m yr$^{-1}$. For the lower part, the ice cliff retreated and flattened under the squeeze of high ice flux input from the upper part, which led to the negative surface elevation change for about $-5$~$-13$ m yr$^{-1}$ (Figure 7f). The evolution of such a cliff-pond system is critically linked to the glacier dynamic process of Zhuxi Glacier.

### 5.2. Comparison with other Glaciers in HMA

The mean surface elevation change of Zhuxi Glacier between 2000 and 2016 was $-0.93 \pm 0.13$ m yr$^{-1}$, which is similar to the mass loss of the debris-covered glaciers nearby ($-0.83 \pm 0.57$ m yr$^{-1}$ during 2000–2014) [32]. Moreover, our results agree with previous geodetic results of $-0.76 \pm 0.19$ m yr$^{-1}$ [35] and $-0.96 \pm 0.38$ m yr$^{-1}$ [65]. Compared with the debris-covered glaciers in the south-slope Himalayas [63,66–69], the mass loss of Zhuxi

Glacier also shows a similar heterogeneous pattern due to the popular existence of the cliff-pond systems. However, the low density of ablation hotspots leads to relatively less mass loss contribution than those representative glaciers in the south-slope Himalayas. The supraglacial ponds and ice cliffs of Zhuxi Glacier only covered ~0.5% total area of the glacier tongue. For the debris-covered glaciers, e.g., Shalbachun Glacier (10.2 km$^2$) and Lirung Glacier (6.5 km$^2$) in Langtang catchment, the thinning rates were $-1.30 \pm 0.20$ m yr$^{-1}$ and $-1.67 \pm 0.59$ m yr$^{-1}$ in 2006–2015, respectively [63]. The supraglacial ponds and ice cliffs of the two glaciers covered about 1.5% to 3.7% area of the glacier tongue [56]. However, similar to other debris-covered glaciers in the Himalayas, Zhuxi Glacier also shows an accelerating trend of mass loss in recent decades due to the anthropogenic warming trend [53,67], which increased from $-0.99 \pm 0.13$ m yr$^{-1}$ during 2000–2016 to $-1.47 \pm 0.25$ m yr$^{-1}$ during 2020–2021 in our UAV survey area.

According to significantly different patterns of surface velocity and surface elevation change, the area below 3400 m a.s.l. for Zhuxi Glacier was classified as a stagnant regime. The stagnant glacier tongue is very common in the Himalayas. Among the 20 debris-covered glaciers in the south-slope Himalayas, 12 glaciers have a stagnant regime of 3~6 km long and have notable surface velocity changes at the upper border of the stagnant regime [70]. The surface velocity of the stagnant regime is commonly under 5 m yr$^{-1}$, whereas the active regime reaches more than 10 m yr$^{-1}$ [71]. In an active regime above 3400 m a.s.l., the high surface velocity greatly influenced the surface elevation change pattern. The area above 3400 m a.s.l. shows an alternating pattern of positive and negative surface elevation change while the average surface elevation change is still negative (Figure 3b). Due to the undulating topography and high surface velocity, this feature is commonly found in many UAV studies on debris-covered glaciers [40,50].

*5.3. Rapid Expansion of Moraine-Dammed Lake and Possible Outburst Risk*

The rapid expansion of the moraine-dammed lake at the terminus is most distinct from the Zhuxi Glacier in the past decade, which is sometimes found on other debris-covered glaciers in the Himalayas [24,25,72,73]. The exponential expansion of the proglacial lake of Zhuxi Glacier experienced two stages (Figure 6). The first stage is the evolution of a supraglacial pond into a proglacial moraine-dammed lake before 2017. When the bottom of the supraglacial pond reached the base of the glacier, it received the meltwater from upstream and became a base-level lake [25,74] instead of pond drainage with the connection of a subglacial conduit [14,75]. The base-level lake of Zhuxi Glacier continued expanding by coalescing other supraglacial ponds and then becoming a proglacial moraine-dammed lake. Moreover, the second stage is the rapid proglacial lake expansion due to the stream channel migration. The proglacial stream channel shifted from the east side to the west side in 2018–2019 (Figure 6a). The input of meltwater contributes to the rapid supraglacial lake expansion and thus contributes to the rapid ice cliff melting and calving [76]. The proglacial lake expanded 6.5 times that in 2017, from 14,841 m$^2$ to 95,790 m$^2$ from 2017 to 2021 (Figure 6b).

Such rapid expansion of proglacial lakes poses the risk of dam failure and may trigger GLOFs [29,77–79]. The dam failures are mainly triggered by some key factors such as the rock/ice avalanche [80,81], the melt of buried ice in the dam [62], and the increased hydrostatic pressure [58]. Catastrophic glacier disasters are an important geomorphic process that may pose a significant threat to communities and infrastructure in high mountains and downstream regions [81]. The ice cliff in hotspot A was the highest ice cliff among adjacent ice cliffs, which was about 30 m high (Figure 7a). This ice cliff calved continuously during the period from 2017 to 2021. Under the calving of the subaqueous ice ramp, the unsupported part of the ice cliff collapses and falls into the lake. The falling part of the ice cliff entering into the lake could generate displacement wave/seiche, which is capable of overtopping the terminal moraine when the impulse wave height is higher than the ~21 m moraine dam (Figure 8d) [62,82]. In addition, the instability of the moraine dam also contributes to the likelihood of GLOF. The proglacial lake of Zhuxi Glacier is dammed by an ice-cored moraine covered with

vegetation. The surface elevation of ice-core moraine has decreased by 15–27 m from 2020 to 2021 (Figure 8a,d). As the ice-cored dam was eroded, the surface topography greatly changed during the past five years, and the proglacial lake significantly expanded northwest downward (Figure 8a,b). Another factor that may trigger the GLOF is the unusual rapid expansion of proglacial lakes or extreme heavy rainfall, which may increase the hydrostatic pressure on the moraine-dammed lake. The expansion of the proglacial lake may merge with other supraglacial ponds, which will suddenly expand the lake area and greatly increase the possibility of dam failure. For exploring the future development of ice cliffs and glacial lakes in hotspots A and B, this study made a topographical transection between these two hotspots (Figure 8c). The south-facing ice cliff in hotspot B retreated ~58 m in 2017–2021, and the ice cliff in hotspot A has retreated ~84 m. The distance between the two hotspots was only ~337 m in 2021. If two ice cliffs continue retreating as the mean rate in 2017–2021, the proglacial moraine-dammed lake of Zhuxi Glacier will coalesce the pond in hotspot B and forms a larger lake favoring potential GLOF within a decade. The outburst of this expanding proglacial lake will greatly threaten the local downstream residents in this valley (149 permanent residents in 2021) and the planned Sichuan–Tibet railway, which is constructed about 8 km down away from the Zhuxi basin.

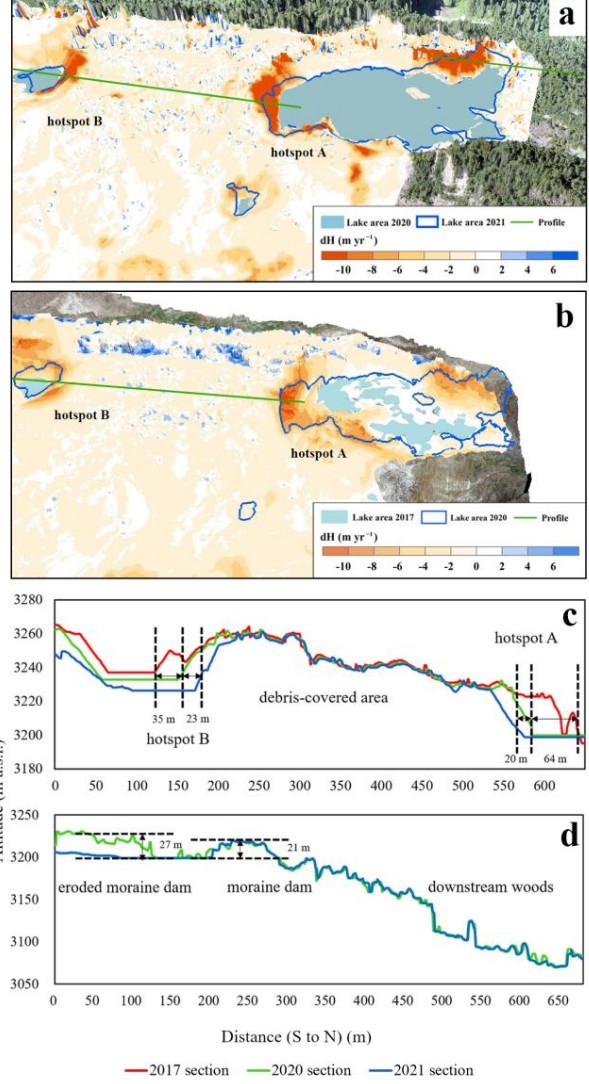

**Figure 8.** The 3-D textured model and surface elevation change of Zhuxi Glacier terminus between 2020 and 2021 (**a**) and 2017–2020 (**b**). The DEM profile along the transection of hotspot A and hotspot B in 2017, 2020, and 2021 (**c**). The DEM profile along the transection from moraine dam to the valley bottom in 2020 and 2021 (**d**).

## 6. Conclusions

In this study, the dynamic change of Zhuxi Glacier, a typical thick debris-covered glacier located in the southeast TP was analyzed in the aspect of surface elevation change, surface velocity, ice cliff, and water bodies by satellite images and high-resolution UAV survey from 2000 to 2021.

Our results showed that the Zhuxi Glacier displayed an accelerating trend of surface lowering in the past 20 years, which increased from $-0.99 \pm 0.13$ m yr$^{-1}$ in 2000–2016 to $-1.47 \pm 0.25$ m yr$^{-1}$ in 2020–2021 within the survey region. The mass loss of this glacier shows high spatial heterogeneity: most of the mass loss occurred on the ablation hotspots, while thick debris-covered areas showed little surface lowering. According to the significantly different spatial patterns of surface elevation change and surface velocity, this glacier was divided into the active regime and stagnant regime along the elevation of 3400 m a.s.l. The mean surface velocity of the active regime was 13.1 m yr$^{-1}$ from 2016 to 2020, which was five times higher than that of the stagnant regime. In addition, the contrasting dynamic condition favors the different hotspots on the Zhuxi Glacier. Three types of hotspots were roughly classified in this study, which represent the persistent cliff-lake system controlled by the proglacial moraine-dammed lake, the expanding cliff-pond system controlled by the aspect of an ice cliff in the stagnant regime, and the shrinking cliff-pond system controlled by glacier dynamic process such as large ice flux in the active regime.

This study also found the rapid expansion of water bodies of Zhuxi Glacier, especially the proglacial moraine-dammed lake, during the past decade. This lake exponentially expanded from 1128 m$^2$ in 2009 to 95,790 m$^2$ in 2021. The expanding lake calved the nearby stagnant ice, destabilized the ice-cored dam, and brought the possibility of sudden ice cliff collapse and lake coalescing, which may bring the risk of dam failure and GLOF, then threaten the residents and infrastructure downstream. A monitoring and early warning system is therefore recommended to be established in the Zhuxi Glacier. Our continuous high-resolution satellite observations (synthetic aperture radar, optical images) and UAV repeated survey will benefit monitoring the dynamic change of cliffs and proglacial lake. The planned in situ and real-time monitoring system (dGPS, water level monitoring, photographs, meteorological and seismic observation) will capture the precursors of ice collapse and glacial lake outburst and raises alarms in advance for downstream communities.

**Author Contributions:** Conceptualization, Z.H. and W.Y.; methodology, Z.H., W.Y. and S.R.; software, Z.H., S.R. and C.L.; validation, Z.H., W.Y. and C.Z.; formal analysis, Z.H. and W.Y.; investigation, Z.H., W.Y., C.Z. and Y.W.; resources, Z.H., W.Y. and C.Z.; data curation, Z.H.; writing—original draft preparation, Z.H.; writing—review and editing, W.Y. and C.Z.; visualization, Z.H. and W.Y.; supervision, W.Y.; project administration, W.Y.; funding acquisition, W.Y. All authors have read and agreed to the published version of the manuscript.

**Funding:** The study is funded by the National Natural Science Foundation of China (41988101, 41961134035) and the National Key Research and Development Project (2019YFC1509102, 2017YFA0603101).

**Data Availability Statement:** The data presented in this study are available on request from the corresponding author.

**Acknowledgments:** We are grateful to the satellite data providers: Planet for Planet Scope Rapideye and Planet Scope One Satellite 4-band, USGS for SRTM-C DEM.

**Conflicts of Interest:** The authors declare no conflict of interest.

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
