# Peer review of "Dynamic Changes of a Thick Debris-Covered Glacier in the Southeastern Tibetan Plateau"

_remotesensing, doi:10.3390/rs15020357_

Round 1

Reviewer 1 Report

Comments on Zhen et al “Dynamic changes of a thick debris-covered glacier in the southeastern Tibetan Plateau” submitted to Remote Sensing

General comments

The dynamic change of debris-covered glacier is critical important for comprehensively understanding the glacier response to recent climate change on the Tibetan Plateau. Based on the repeated high-resolution UAV surveys and remote sensing images, Zhen et al. investigated the dynamic changes (surface elevation, surface velocity, supraglacial lakes, and ice cliffs) of Zhuxi Glacier, a typical thick debris-covered glaciers in the southeastern Tibetan Plateau. This study found the accelerating mass loss trend and the contrasting active/stagnant pattern of this debris-covered glacier. And the surface characteristics including ice cliff, supraglacial lakes have also changed greatly in response to recent glacier melting. Particularly, the exponential expansion of the proglacial lake and the significant downwasting of stagnant ice near the dammed terminus moraine possibly trigger the GLOF. Overall, this paper is generally well-written and helpful for understanding the dynamic change of the representative debris-covered glaciers in the monsoon influenced Tibetan Plateau. However, several points should be considered in the revised manuscript before publication.

Major comments

The authors outlined major findings of the study in the abstract, but it seems those findings are irrelevant among those findings and just list out them, so I suggest a well reorganized abstract.  Line 21-23, ”The surface velocity analysis 18 indicated that the whole glacier can be divided into an active regime and a stagnant regime along 19 the elevation of 3400 m a.s.l. The active regime moved downward with a mean surface velocity of 20 13.1 m yr−1, which was five times higher than that of the stagnant regime.” The sentence is hard to understand.

Besides, this study analyzed surface change of debris-covered and debris free, ice velocity, ice cliff ablation and supraglacial ponds, they are irrelevant between each other and lack of a unifying theme in the section of introduction. I suggest a well-organized introduction, that could connect those topics to a scientific problem.

Minor comments

1. The representatives of the studied glacier. Zhuxi Glacier is one of thick debris-covered glaciers in the southeastern Tibetan Plateau. The author should explain the representatives of Zhuxi Glacier in this region in the introduction or Study sites. Why did you select Zhuxi Glacier as the case study? Unlike the Hailuogou glacier (Zhang et al. 2011) and the 24K Glacier (Yang et al., 2017), the debris of Zhuxi Glacier is very thicker. Are there many thick debris-covered glaciers, like Zhuxi Glacier in the this monsoon-influenced region? Recent studies have provide the debris thickness distribution in the world. The author should check the debris thickness nearby glaciers and addressed the representatives of Zhuxi Glacier in this region.

2. Comparison with other geodetic studies. This study provide the mean mass loss of Zhuxi Glacier during the period between 2000 to 2016 by comparing SRTM DEM and ZY-DEM. Indeed, Recent studies (e.g. Brun et al., 2018; Shean et al., 2020; Hugonnet et al., 2021) have provide the regional mass loss on the whole Tibetan Plateau. The comparison with these geodetic studies is necessary to validate the reliability of your result based on satellite-derived DEM difference. Such comparison should be added in the Section “5.2 Comparison with other glaciers in HMA”

3. The dividing surface velocity between active and stagnant ice. This study used an average surface velocity of 7 m yr-1 to distinguish the active zone and stagnant zone for the Zhuxi Glacier. However, in the Section 5.2, “The surface velocity of stagnant regime is commonly under 5 m yr-1 ” as the threshold value. You should provide the explain why different threshold values used between the Everest glaciers and Zhuxi Glacier for disgusting their dynamics.

4. The evolution of water bodies. The exponential expansion of the proglacial lake dominate the overall lake area change for the Zhuxi Glacier. If the proglacial lake was excluded, the area change maybe display insignificant trend during the past decade. Therefore, in Figure 5b, the area change without the proglacial lake is suggested to be added. And why did the number of supraglacial lakes decrease to be about 15-17 in 2015 and 2016? Could you explain such abnormal change?  In addition, the possible threshold (the minimum area or width/length) for detecting the supraglacial lakes should be introduced.

5. Possible outburst risk. The authors have carried out three repeated UAV surveys in 2017, 2020 and 2021. However, in Figure 8a, the author only provided the elevation change during 2020-2021. Could you add the elevation change of 2017-2020 for the this key region? This new figure is helpful for demonstrating the rapid change of ice cliffs and supraglacial lakes. In Figure 8b, the elevation profile of the dammed moraine and the down valley is suggested to be included for showing their topography contrasting between supraglacial lake and valley bottom.

6. The English should be polished by a native speaker.

7. I suggest the UAV data should be available for the public.

8. The style of Tables and Figures should be following the MDPI Style Guide. e.g. The panels with more than one part was labeled a, b, c, d but with different front size.

Reviewer 2 Report

The reviewer would like to thank the authors for this thoughtful manuscript. This work has good potential. The authors are requested to put in some additional efforts to improve the quality of this manuscript. 

Introduction 

The authors are requested to discuss and cite an important article reporting a glacier related disaster over the Himalayas. The discussion must be in the context of the significance of the dynamics of the debris covered glaciers.

-Shugar et al, A massive rock and ice avalanche caused the 2021 disaster at Chamoli, Indian Himalaya, Science, 2021

Active and Stagnant Regime

The authors have reported a large difference in the surface velocity between the active and stagnant regime. Please explain how this is related to the crevassing and formation of surface features like drainage ponds and their population.

SAR Sensor for Cryospheric Monitoring

The authors are requested to discuss the following articles on the possibility of including satellite based SAR sensors for monitoring cryospheric entities like snow fields and glaciers. 

-Muhuri et al., “Snow cover mapping using polarization fraction variation with temporal RADARSAT-2 C-band full-polarimetric SAR data over the Indian Himalayas”, IEEE JSTARS, 2018.

-Tsai, Y.L.S., et al., 2019. Remote Sensing of Snow Cover Using Spaceborne SAR: A Review. Remote Sensing.

Figures

The authors are requested to provide figures with a higher resolution. The information presented in the figure is not very clear like the elevation (Fig. 5) info over the glacier. Some more work is requested. In Fig. 3 the location of the velocity info presented is not clear over the glacier. Are you missing a transect over the glacier? 

Conclusion 

The authors are requested to list the key contributions in this section. At the moment the section is not detailed enough.
